# High risk *APOL1* genotypes and kidney disease among treatment naïve HIV patients at Kano, Nigeria

**Aliyu Abdu**[1]*, **Raquel Duarte**[2], **Caroline Dickens**[2], **Therese Dix-Peek**[2], **Sunusi M. Bala**[3], **Babatunde Ademola**[1], **Saraladevi Naicker**[2]

**1** Department of Medicine Aminu Kano Teaching Hospital/ Bayero University Kano, Kano, Nigeria, **2** Department of Internal Medicine, Faculty of Health Sciences, University of the Witwatersrand, Johannesburg, South Africa, **3** Department of Medicine, M.A. Wase Teaching Hospital, Kano, Nigeria

* aliyuabdu2000@yahoo.co.uk

## Abstract

### Introduction

Racial disparities are known in the occurrence of kidney disease with excess risks found among people of African descent. *Apolipoprotein L1* (*APOL1*) gene variants G1 and G2 are associated with kidney disease among HIV infected individuals of African descent in the USA as well as among black population in South Africa. We set out to investigate the prevalence of these high-risk variants and their effects on kidney disease among HIV infected patients in Northern Nigeria with hitherto limited information despite earlier reports of high population frequencies of these alleles from the Southern part of the country.

### Methods

DNA samples obtained from the whole blood of 142 participants were genotyped for *APOL1* G1 and G2 variants after initial baseline investigations including assessment of kidney function. Participants comprised 50 HIV positive patients with no evidence of kidney disease, 52 HIV negative individuals with no kidney disease and 40 HIV positive patients with chronic kidney disease (CKD) evidenced by persistent proteinuria and/or reduced eGFR, who also had a kidney biopsy. All the HIV positive patients were newly diagnosed and treatment naïve.

### Results

The distribution of the APOL1 genotypes among the study participants revealed that 24.6% had a G1 risk allele and 19.0% a G2. The frequency of the High Risk Genotype (HRG) was 12.5% among those with CKD compared to 5.8% in the HIV negative group and zero in the HIV positive no CKD group. Having the HRG was associated with a higher odds for developing HIV Associated Nephropathy (HIVAN) (2 vs 0 risk alleles: OR 10.83, 95% CI 1.38–84.52; P = 0.023; 2 vs 0 or 1 risk alleles: OR 5.5, 95% CI 0.83–36.29; P = 0.07). The HRG was also associated with higher odds for Focal Segmental Glomerulosclerosis (FSGS) (2 vs

**Data Availability Statement:** All relevant data are within the paper and its Supporting Information files.

**Funding:** The Authors received no specific funding for this work

**Competing interests:** NO authors have no competing interests

0 risk alleles: OR 13.0, 95% CI 2.06–81.91; P = 0.006 and 2 vs 0 or 1 risk alleles: OR 9.0, 95%CI 1.62–50.12; P = 0.01) when compared to the control group.

## Conclusion

This study showed a high population frequency of the individual risk alleles of the *APOL1* gene with higher frequencies noted among HIV positive patients with kidney disease. There is high association with the presence of kidney disease and especially FSGS and HIVAN among treatment naive HIV patients carrying two copies of the HRG.

## Introduction

There is global disparity in the occurrence of kidney disease with an excess risk for persons of African Ancestry as they were shown to have four times higher incidence of chronic kidney disease (CKD) compared to people of European descent [1], together with a more rapidly progressive course of the disease, as reported in the Multi-Ethnic Study of Atherosclerosis (MESA) during 5 years of follow-up [2]. Data analysis from the US revealed that people of African descent showed a consistent yearly increase in the incidence of End Stage Kidney Disease (ESKD) that is 3.5 to 5-fold higher than people of European descent [3]. This racial/ ethnic disparity in the occurrence of kidney disease is also noted among HIV infected patients. Eggers et al reported that the possibility of developing kidney disease among HIV infected people of African descent is at least 50 times higher than that of people of European descent [4]. Nearly 90% of ESKD attributed to HIV-Associated Nephropathy (HIVAN) was likely to occur in persons of African ancestry [5].

The excess risk of kidney disease among people of African descent could not entirely be explained by factors such as access to care and other socioeconomic factors [6]. The possibility of genetic factors being significant in the development of kidney disease among this population group was supported by findings from many studies [7, 8]. Variants in the gene located on chromosome 22 that encode for *Apolipoprotein L1* (*APOL1*) have been found to be strongly associated with increased risk of various forms of kidney diseases among people of African descent, with three times greater risk of developing lupus nephritis, seven times greater risk of developing hypertensive nephrosclerosis, seventeen times greater risk of developing primary Focal Segmental Glomerulosclerosis (FSGS) and twenty nine times greater risk to develop HIVAN [9, 10]. There were reports of even stronger associations with the development of HIVAN [11, 12]. Kasembeli et al, in a study among black people in South Africa, showed that the presence of these high risk variants was associated with an 89-fold odds for developing HIVAN [13].

These alleles include the *APOL1* G1 variant comprising two non-synonymous coding variants rs73885319 and rs60910145 and the G2 allele with a 6 base pair deletion that removes two amino acids (rs71785313).

*APOL1* high-risk genotypes (HRGs) are defined as the occurrence of two risk alleles in any of the combinations (homozygous G1/G1, homozygous G2/G2 and compound heterozygous G1/G2). Patients with the HRG carry an increased risk of kidney disease. The heterozygous forms (G1/G0 or G2/G0), with only one risk allele, are known as low risk genotypes and are associated with a lesser risk of kidney disease. These kidney disease risk variants occur in higher frequency among individuals of West African descent [14]. This was later found to have evolutionary origins by way of natural selection for a trait that offers protection against

infection with *Trypanosoma brucei*, a protozoan that causes sleeping sickness, which is endemic in parts of Sub-Saharan Africa. These traits encode for circulating APOL1 that functions as a trypanolytic factor leading to death of the trypanosome parasites in human serum [15]. However this positive selective advantage against sleeping sickness has associated increased risk of developing CKD among the carriers of the HRG.

Genetic predilection to HIVAN has been described among people of African descent but has not been widely studied among native Africans with the study among the South African black population by Kasembeli et al and few other studies from Nigeria as well as a study among children in the Democratic Republic of Congo (DRC) [13, 16–18]. We hypothesised that the frequency of *APOL 1* high risk genotypes will be high, with higher odds for kidney disease among HIV infected treatment naive patients in Kano. We therefore set out to investigate the frequencies of these variants and their association with kidney disease among HIV infected persons from this part of Northern Nigeria, a region with scarce information regarding these risk alleles despite previous reports of high population frequencies of these alleles from the South Western and South Eastern parts of the country which were predominantly inhabited by the Yoruba and Igbo population groups respectively [14, 19].

## Methods

Participants in this cross-sectional study were recruited consecutively from the dedicated HIV clinic of Aminu Kano Teaching Hospital (AKTH), one of the approved United States President's Emergency Plan for AIDS Relief (PEPFAR) centers in Northern Nigeria. One hundred and fifty five participants comprising 50 treatment naive HIV positive patients with no evidence of kidney disease (HIV positive no CKD), 52 HIV negative participants with no kidney disease (HIV negative no CKD) and 53 HIV positive patients with kidney disease evidenced by persistent proteinuria and/or reduced estmated glomerular filtration rate (eGFR), who had a kidney biopsy (HIV positive CKD) also. The HIV positive CKD patients had no obvious other known causes of kidney disease, such as diabetes and hypertension, and were not being treated for any kidney disease prior to enrolment and had no contraindication to kidney biopsy. The HIV negative with no kidney disease were recruited from among volunteer hospital workers and patients at general out patients who presented with minor ailments. Patients were recruited into the study between April 2016 and March 2018.

A spring-loaded automatic biopsy gun (BARD 16/18 G, 22 mm edge) was used to perform percutaneous kidney biopsy under real-time ultrasound guidance as a daytime procedure. Before the biopsy, the patients were counseled on the proceedure, had a complete blood count including platelet measurement. The coagulation profile was also checked to rule out any bleeding tendency, and an informed consent for the proceedure was obtained. The patients were monitored before and after the biopsy according to the established unit protocol and were discharged after at least 8 hours of observation and no obvious complications observed.

All participants gave a written informed consent to participate and underwent an initial assessment with a focused clinical examination and baseline investigations including serum creatinine, CD4 count, complete blood count, Hepatitis B surface antigen and Hepatitis C antibody tests. The IDMS traceable Jaffe method was used to determine creatinine levels in both the blood and urine samples using the Cobas C 311 Clinical Chemistry Analyser (Roche Diagnostics Rotkreuz, Switzerland) at Aminu Kano Teaching Hospital (AKTH). The eGFR was calculated using the Chronic Kidney Disease Epidemiology Collaboration (CKD-EPI) equation without use of the ethnicity factor.

Genomic DNA was extracted from whole blood using the Maxwell DNA purification kit (Promega AS1010, USA) as per the manufacturer's protocol. DNA concentrations were

determined by the NanoDropTM 2000 spectrophotometer (Thermo Scientific, USA) and the quality was assessed by means of the A260/280 ratios. A 260/280 ratio of 1.8 to 2.0 was considered as indicative of a pure DNA preparation free of contamination. DNA samples were genotyped for SNPs rs73885319, rs60910145 and rs71785313 using a PCR-restriction fragment length polymorphism assay as described by Nqebelele et al. [20] Genomic studies were carried out in the Department of Internal Medicine Research Laboratory at the University of Witwatersrand, Johannesburg.

Ethical approval to conduct the study was granted by the Research Ethics Committee of AKTH (NHREC/21/08/AKTH/1090) and University of Witwatersrand Human Research Ethics Committee (M140116).

## Statistics

Data was analysed using STATA v14.2 (Stata Corp, College Station, TX). Demographic and clinical characteristics were expressed as mean ± SD. Comparisons of continuous variables were made using a Kruskal-Wallis test or ANOVA test, as appropriate. Categorical variables were expressed as frequencies and compared using a chi-squared test. Associations between the risk of kidney disease and *APOL1* risk alleles were explored using logistic regression models. A two-tailed p-value $<0.05$ was considered significant. The SNP data was tested for any departure from Hardy-Weinberg equilibrium.

## Results

A total of 155 participants were studied with a mean age of 36.1 ± 9.3years and 55% were males. Complete data on genotyping was available for 142 participants (40 HIV positive CKD patients, 50 HIV positive controls and 52 HIV negative controls). Table 1 shows the characteristics of the participants. The HIV positive CKD patients were older than the HIV positive no-CKD patients (38.8 ± 9.3 vs. 31.2 ± 7.4 years, P<0.001); however the age was similar to the HIV negative no-CKD controls (38.8 ± 9.3 vs 38.0 ± 9.3, P = 0.648). The HIV positive CKD patients also had higher blood pressure (systolic: 120 vs 110 mm Hg, P = 0.001) when compared with the no-CKD controls, both HIV positive and negative. All the HIV positive CKD patients had a kidney biopsy. The histology results revealed that HIVAN (characterised by collapsing FSGS, cystic tubular dilatation, interstitial infiltrates), was reported in 17 (32%) of the patients. Majority of the patients had FSGS [20 (37.7%)] and HIVAN [32% (17)] as their histologic diagnosis while 6 (11.3%) had no significant pathological finding, 7(13.2%) had chronic interstitial nephritis, and acute tubular necrosis and membranoproliferative

Table 1. Characteristics of the study participants.

| Characteristic | HIV CKD (n = 53) | HIV positive controls(n = 50) | HIV negative controls (n = 52) | P-value |
|---|---|---|---|---|
| Male, n (%) | 34 (64.2%) | 20 (40.0%) | 32 (61.5%) | 0.027 |
| Age (years) | 38.8 ± 9.3 | 31.2 ± 7.4 | 38.0 ± 9.3 | 0.083 |
| Serum creatinine (μmol/L) | 210 (143–302) | 82 (70–90) | 82 (73–87.5) | <0.001 |
| eGFR (ml/min/1.73m$^2$) | 34.4 (26.4–48.9) | 107 (87–128) | 110.5 (100.5–120.5) | <0.001 |
| Systolic BP (mmHg) | 120.0 (110–132) | 110.0 (110–120) | 110.0 (110–120) | 0.001 |
| Diastolic BP (mmHg) | 77.0 (65–84) | 60.0 (60–70) | 69.5 (60–70) | <0.001 |
| Body mass index (kg/m$^2$) | 22.26 ± 2.6 | 22.5 ± 7.7 | 21.9 ± 6.6 | 0.770 |
| CD4 (cells/mm3) | 115 (74–304) | 265 (179–402) | - | 0.005 |

HIV = Human Immune deficiency Virus, CKD = Chronic Kidney Disease, eGFR = estimated glomerular filtration rate, BP = blood pressure; n = number

glomerulonephritis were reported in one patient each. The majority of the FSGS seen in this study were of the NOS sub type with only a few being of the tipped sub type. Among the patients that had immunofluorescence evaluation of their tissues (n = 10), none showed features of immune complex disease.

The frequency of *APOL 1* genotypes showed that 24.6% (35/142) of the study participants had at least 1 G1 risk allele and 19.0% (27/142) had a G2 risk allele. The haplotype G-G-I (G1$^{GM}$), comprising the derived allele at both rs73885319 (p.S342G) and rs60910145 (p. I384M) and the ancestral insertion (I) allele at rs71785313, was the most frequent G1 configuration seen in 20.4% of all the study participants. Table 2 depicts the distribution of the various haplotypes across different groups of study participants.

The alleles of the ancestral haplotype G0 include; The G1$^{GM}$ haplotype with two missense alleles; the G1$^{G+}$ haplotype with one missense risk allele at rs73885319; the G1$^{+M}$ has one missense variant at rs60910145. The G2 haplotype has the 6 base pair deletion risk allele at rs71785313.

The frequency of the HRG was 5.6% (8/142) among all the study participants. Three participants had the homozygous G1/G1 genotype and five had the compound heterozygote G1/G2 form; none of the study participants had the homozygous G2/G2 form. However the frequency of the HRG was 12.5% among those with HIV CKD compared to 5.8% in the HIV negative controls and zero in HIV positive no-CKD group; this difference was statistically significant (p = 0.025). Table 3 compares the distribution of the risk alleles among the various study groups.

We found a 10.8 fold higher odds for HIVAN among those with 2 risk alleles vs no risk alleles (OR 10.83, 95%CI 1.38–84.52; P = 0.023) and a 5.5 fold higher odds for those with 2 vs 1or 0 risk alleles (OR 5.50, 95%CI 0.83–36.29; P = 0.07) compared to controls. Associations were stronger among those with FSGS (2 vs 0 risk alleles: OR 13.0, 95%CI 2.06–81.91;

**Table 2. Distribution of *APOL1* SNP genotypes and haplotypes in the study participants with HIV positive CKD, HIV positive no-CKD and HIV negative no-CKD.**

| *APOL 1* variants | HIV Positive CKD (n = 40) | HIV Positive no-CKD (n = 50) | HIV Negative no-CKD controls (n = 52) | P-value | P-value HIV+ CKD vs HIV+ no-CKD | P-value HIV+ CKD vs HIV- no-CKD |
|---|---|---|---|---|---|---|
| **rs73885319** | | | | | | |
| AA, n (%) | 32 (80.0%) | 43 (89.0%) | 36 (69.2%) | 0.008 | 0.143 | 0.023 |
| AG, n (%) | 5 (12.5%) | 7 (14.0%) | 16 (30.8%) | | | |
| GG, n (%) | 3 (7.5%) | 0 | 0 | | | |
| **rs60910145** | | | | | | |
| TT, n (%) | 30 (75.0%) | 43 (86.0%) | 36 (69.2%) | 0.153 | 0.289 | 0.373 |
| TG, n (%) | 9 (22.5%) | 7 (14.0%) | 16 (30.8%) | | | |
| GG, n (%) | 1 (2.5%) | 0 | 0 | | | |
| **rs71785313** | | | | | | |
| Insertion/Insertion | 28 (70.0%) | 46 (92.0%) | 41 (78.8%) | 0.027 | 0.007 | 0.331 |
| Insertion/Deletion | 12 (30.0%) | 4 (8.0%) | 11 (21.2%) | | | |
| Deletion/Deletion | 0 | 0 | 0 | | | |
| **Haplotypes** | | | | | | |
| G0 | 35 | 50 | 49 | | | |
| G1$^{GM}$ | 8 | 6 | 15 | | | |
| G1$^{G+}$ | 2 | 1 | 1 | | | |
| G1$^{+M}$ | 2 | 1 | 1 | | | |
| G2 | 12 | 4 | 11 | | | |

HIV = Human Immune deficiency Virus, CKD = Chronic Kidney Disease.

**Table 3. *APOL1* genotype distribution among HIV positive CKD patients and control patients (HIV positive no-CKD and HIV-negative no-CKD).**

| Characteristic | HIV positive CKD (n = 40) | HIV positive no-CKD controls (n = 50) | HIV-negative no-CKD controls (n = 52) | p-value | p-value HIV+ CKD vs HIV+ no-CKD | p-value HIV+ CKD vs HIV- no-CKD |
|---|---|---|---|---|---|---|
| **0 risk alleles** | | | | | | |
| G0/G0 | 20 (50.0%) | 38 (76.0%) | 27 (51.9%) | 0.013 | 0.015 | 1.000 |
| **1 risk allele** | | | | | | |
| G0/G1 | 5 (12.5%) | 8 (16.0%) | 14 (26.9%) | 0.13 | 0.175 | 0.673 |
| G0/G2 | 10 (25.0%) | 4 (8.0%) | 8 (15.4%) | | | |
| **2 risk alleles** | | | | | | |
| G1/G1 | 3 (7.5%) | 0 | 0 | 0.025 | 0.015 | 0.288 |
| G1/G2 | 2 (5.0%) | 0 | 3 (5.8%) | | | |
| G2/G2 | 0 | 0 | 0 | | | |

Data is represented as n (%) HIV = Human Immune deficiency Virus, CKD = Chronic Kidney Disease,

P = 0.006 and 2 vs 0 or 1 risk alleles: OR 9.0, 95% CI 1.62–50.12; P = 0.012) and remained significant after adjusting for age and gender. Table 4 depicts the OR in the various groups, both unadjusted and after adjustments for age and gender.

## Discussion

This study conducted in Northern Nigeria determined the frequencies of *APOL1* genotypes among treatment naïve HIV infected patients with biopsy-proven kidney disease and those with no evidence of kidney disease.

The prevalence of the haplotypes G1 (24.6%) and G2 (19.0%) in this population was lower than that reported in previous studies involving Nigerian populations; 45 to 50% for G1 among Yoruba population in the South Western parts of Nigeria and 30% among the Igbo population in the South Eastern parts of the country [12, 14, 19, 21]. This finding is also lower than that of our previous report, which comprised of a larger cohort with a multi-ethnic population of treatment experienced HIV patients [17]. It is also lower than the 40% reported for the Ashanti population and higher than the 11.4% reported among the Bulsa population, both from Ghana [20]. The G1 prevalence in our study participants was similar to the 20–22%

**Table 4. Odds ratios for associations of *APOL1* risk alleles with HIVAN and FSGS versus controls.**

| | Unadjusted OR (95% CI) | P-value | Adjusted OR* (95% CI) | P-value |
|---|---|---|---|---|
| **HIVAN** | | | | |
| No of Risk Alleles | | | | |
| 1 vs 0 | 3.82 (1.07–13.61) | 0.038 | 2.51 (0.63–9.92) | 0.189 |
| 2 vs 0 | 10.83 (1.38–84.52) | 0.023 | 4.81 (0.52–44.90) | 0.168 |
| 2 vs 1 | 2.83 (0.40–19.87) | 0.295 | 2.58 (0.36–18.63) | 0.347 |
| 2 vs 1 or 0 | 5.50 (0.83–36.29) | 0.077 | 3.76 (0.54–26.35) | 0.182 |
| **FSGS** | | | | |
| No of Risk Alleles | | | | |
| 1 vs 0 | 2.29 (0.65–8.07) | 0.196 | 1.55 (0.40–6.00) | 0.526 |
| 2 vs 0 | 13.0 (2.06–81.91) | 0.006 | 14.00 (1.77–110.67) | 0.012 |
| 2 vs 1 | 5.67 (0.92–34.99) | 0.062 | 5.30 (0.83–33.95) | 0.079 |
| 2 vs 1 or 0 | 9.0 (1.62–50.12) | 0.012 | 8.85 (1.55–50.74) | 0.014 |

*for age, gender. HIV = Human Immune deficiency Virus, CKD = Chronic Kidney Disease,

reported from 3 surveys among populations of African descent [10, 22]. The prevalence was however higher than the 7.3% for G1 and 11.1% for G2 reported in a South African black population [13]. The G2 prevalence of 19% in this study was higher than the 7.5% and 16.6% reported in the Yoruba population [14, 22]. The G2 frequency in this study was lower than the 24.4% in the Igbo and 21.4% in the Bulsa in Ghana. It was however similar to the 13–15% among people of African descents in the US and the 12.9% among the Asante population in Ghana [10, 21]. Variations in the frequencies of these haplotypes among diverse African populations were also reported by Tzur et al when they studied 676 samples from 12 African populations that included Cameroon (2 population groups), Congo, Ethiopia (4 population groups), Ghana (2 population groups), Malawi, Mozambique and Sudan. The haplotype distribution differed between groups from the same country and also across the continent; for example in the Bulsa and Asante populations of Ghana, the frequencies were 11% and 41% respectively [21]. The findings in this study also differ from that of an Ethiopian population where they reported zero frequencies of these *APOL 1* risk alleles among HIV infected patients with associated absence of HIVAN [23].

From the lower frequencies of these haplotypes in our study compared to those from the southern part, it may be possible to speculate on a possible relaxation of the natural selection push exerted by the Trypanosome infection in these regions by ancestors of the ethnic populations in this Northern part of Nigeria as previously suggested [17]. The study area is some distance from the Trypanosome infection belt and the Atlantic coastal areas of West Africa, the hub for the trans-Atlantic slave trade from which most people of African descent trace their ancestral journeys to the Caribbean and the Americas [24]. The fact that this is a hospital-based study as against a community based survey may be associated with recruitment bias and may also explain the discrepancies observed. There is a need for a multicenter comparative study with a large sample size to explain the differences in the frequencies of these alleles in different geographic locations within the same country. The frequencies of the HRG defined by the presence of 2 risk alleles (G1/G1, G2/G2, or G1/G2) among all participants in this study was 5.6% and it was 12.5% among the HIV positive CKD group. This overall lower frequency was similar to a previous report from this center among ART experienced cohort, although lower than that reported from the southern part of the country [18] and lower than the 11% reported for people of African descent [7] and 7% reported among DRC children [9]. However the higher frequency of the HRG among those with kidney disease was also reported by Kasembeli et al in which the frequency of the HRG was 17% among all participants but was as high as 89% among those with HIVAN [13] and the report by Ekrikpo et al that showed overall higher frequencies of the HRG of more than 70% in the study population with even much higher frequencies among HIV patients with kidney disease [18].

This study found that having the 2 high risk allele genotype was a predictor of FSGS compared to no risk alleles. The association was maintained in the recessive form (2 vs 1 or 0 risk alleles:). The association was also strong with HIVAN (2 vs 0 risk alleles) although the association did not reach statistical significance in the recessive form (2 vs 1 or 0 risk alleles). Kopp et al also reported this association in a cohort of FSGS and HIVAN; a greater association was observed among HIVAN compared to FSGS [10] Kasembeli et al reported an even greater association with HIVAN among a black South African cohort but no significant association with FSGS [13]. The lower strength of the association among our patients may be due to a smaller sample size compared to both the Kopp and Kasembeli studies. This calls for further study with a larger sample size on the effect of this HRG and other *APOL1* nephropathies, not only among HIV infected patients but also in the general population.

This study provides information on the high frequencies of the *APOL1* genotypes among treatment naïve HIV infected patients from this region, though the frequencies were not as

high as those reported among population groups in the western and eastern parts of Nigeria. The study also confirmed the association between the occurrence of these HRGs and kidney disease, and specifically FSGS and HIVAN histological types among HIV infected patients despite the relatively lower frequencies; this was hitherto not reported from this part of the country This calls for the inclusion of evaluation of kidney function among newly diagnosed HIV patients as recommended [25], to allow for early detection and subsequent initiation of ART as extrapolation from a previous study showed that the life time risk of developing HIVAN among HIV patients carrying 2 high risk alleles may reach as high as 50% [10]. This finding also may be of significance considering the multiple risk factors for kidney disease among HIV infected individuals. The strong association with APOL1 HRG may provide an additional strategy in the evaluation and management of kidney disease risk among these patients. This study is limited by a small sample size and cross sectional design that did not allow for the assessment of the impact of the HRG on the progression of the kidney disease. However the patients being ART naïve and having a definitive diagnosis of CKD with kidney histology adds to the strength of the association of HRG of *APOL1* with specific forms of kidney disease among the study patients.

## Supporting information

**S1 Data.**
(XLSX)

## Author Contributions

**Conceptualization:** Aliyu Abdu, Saraladevi Naicker.

**Data curation:** Aliyu Abdu, Raquel Duarte, Caroline Dickens, Sunusi M. Bala, Saraladevi Naicker.

**Formal analysis:** Aliyu Abdu, Caroline Dickens, Therese Dix-Peek.

**Investigation:** Aliyu Abdu, Raquel Duarte, Therese Dix-Peek, Sunusi M. Bala, Babatunde Ademola.

**Methodology:** Aliyu Abdu, Raquel Duarte, Caroline Dickens, Therese Dix-Peek.

**Project administration:** Babatunde Ademola.

**Supervision:** Raquel Duarte, Saraladevi Naicker.

**Writing – original draft:** Aliyu Abdu.

**Writing – review & editing:** Aliyu Abdu, Raquel Duarte, Caroline Dickens, Sunusi M. Bala, Babatunde Ademola, Saraladevi Naicker.

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
