## [Decision Letter · Decision Letter 0]

9 Jul 2021

PONE-D-21-10394

HIGH RISK APOL1 GENOTYPES AND KIDNEY DISEASE AMONG TREATMENT NAÏVE HIV PATIENTS AT KANO, NIGERIA

PLOS ONE

Dear Dr. Aliyu Abdu, 

Thank you for submitting your manuscript to PLOS ONE. After careful consideration, we feel that it has merit but does not fully meet PLOS ONE’s publication criteria as it currently stands. Therefore, we invite you to submit a revised version of the manuscript that addresses the all the points raised  by 3 experts in the fields, which I agreed. Please respond in a point-by-point manor, and please pay special attention to the concern raised by reviewer #1 and #2. Please revise your manuscript carefully, as this revision suggestion may not necessarily lead to publication.

We look forward to receiving your revised manuscript.

Kind regards,

Weijing He, M.D.

Academic Editor

PLOS ONE

Journal Requirements:

2. For studies involving humans categorized by race/ethnicity, age, disease/disabilities, religion, sex/gender, sexual orientation, or other socially constructed groupings, authors should:

1) Explicitly describe their methods of categorizing human populations,

2) Define categories in as much detail as the study protocol allows,

3) Justify their choices of definitions and categories,

4) Explain whether (and if so, how) they controlled for confounding variables such as socioeconomic status, nutrition, environmental exposures, or similar factors in their analysis, and

5) Update outmoded terms and potentially stigmatizing labels to more current, acceptable terminology.

Examples: “Caucasian” should be changed to “white” or “of [Western] European descent” (as appropriate); “XXX victims” should be changed to “patients with XXX.

3. Please provide the recruitment date range in your methods section.

4. Thank you for stating in your methods "All participants gave consent to participate." Please provide additional details regarding participant consent. In the ethics statement in the Methods and online submission information, please ensure that you have specified:

  - whether consent was informed

 - what type of consent you obtained (for instance, written or verbal, and if verbal, how it was documented and witnessed).

7. Please include a copy of Table 4 which you refer to in your text on page 10.

8. We note you have included two tables which you refer in the text of your manuscript, however both are labelled as Table 3. Please ensure that you label each Table by a separate number in the title and also cite the relevant table number in your text; if accepted, production will need this reference to link the reader to each Table.

Reviewers' comments:

Reviewer's Responses to Questions

**Comments to the Author**

1. Is the manuscript technically sound, and do the data support the conclusions?

Reviewer #1: Partly

Reviewer #2: Yes

Reviewer #3: Yes

2. Has the statistical analysis been performed appropriately and rigorously? 

Reviewer #1: Yes

Reviewer #2: Yes

Reviewer #3: Yes

3. Have the authors made all data underlying the findings in their manuscript fully available?

Reviewer #1: Yes

Reviewer #2: Yes

Reviewer #3: Yes

4. Is the manuscript presented in an intelligible fashion and written in standard English?

Reviewer #1: No

Reviewer #2: Yes

Reviewer #3: Yes

5. Review Comments to the Author

Reviewer #1: General Comments:

The study was performed to determine the frequency of APOL1 gene frequencies in HIV-1 infected- ART naïve patients at KANO, the northern part of Nigeria. 142 samples having HIV negative and HIV positive (with or without chronic kidney disease)) were analyzed for risk alleles by PCR-RFLP assay. The authors found an association between risk alleles (G1 and G2) frequencies and CKD. While these observations in part confirm previous studies, the APOL1 gene frequencies at the KANO region

remain to be determined. Simultaneously, the obtained results with respect to published results do not give any clear picture of the association between the tested variables. Consider increasing the sample size, which might help draw definitive conclusions, especially when a broad spectrum of frequency associations is reported. Overall, the data appear to be essential to the complex area of HIV-1 therapy.

Specific Comments:

1. I would ask the authors to include line and page numbers in all future submissions as it makes reviewing difficult without these.

2. Consider building a hypothesis in the introduction

3. Table 2 was mentioned two times and did not come complete on paper when printed.

4. Consider describing Table legends.

Reviewer #2: The authors have presented data on a limited sample size on the prevalence of APOL-1 genotypes in HIV positive patients (with and without kidney disease) in Northern Nigeria, West Africa. The describe the prevalence of high-risk and low-risk genotypes on kidney disease in the region.

I have a number of concerns about the study:

Major concerns:

1. The aim of the study was to “investigate the prevalence of high-risk variants and their effects on kidney disease among HIV infected patients in Northern Nigeria with hitherto limited information despite earlier reports of high population frequencies of these alleles from the Southern part of the country.” There are a number of prevalence studies from the same country region including this one from the same centre with a much larger sample size:

(Apolipoprotein-1 risk variants and associated kidney phenotypes in an adult HIV cohort in Nigeria. Wudil UJ, Aliyu MH, Prigmore HL, Ingles DJ, Ahonkhai AA, Musa BM, Muhammad H, Sani MU, Nalado AM, Abdu A, Abdussalam K, Shepherd BE, Dankishiya FS, Burgner AM, Ikizler TA, Wyatt CM, Kopp JB, Kimmel PL, Winkler CA, Wester CW.Kidney Int. 2021 Apr 23:S0085-2538(21)00430-0. doi: 10.1016/j.kint.2021.03.038. Online ahead of print.).

So, it is unclear what added value this study has provided. Rather, I think their results are conflicting with recent results from the same centre.

2. Leading on to the study rationale was this statement “Genetic predilection to HIVAN has been described among African-Americans but has not been widely studied among native Africans except for a few studies such as the study among the South African black population by Kasembeli et al and another study among children in the DRC.” This is not correct as there have been several other studies e.g.:

- Association of Genetic Polymorphisms of TGF-β1, HMOX1, and APOL1 With CKD in Nigerian Patients With and Without HIV. Ekrikpo UE, Mnika K, Effa EE, Ajayi SO, Okwuonu C, Waziri B, Bello A, Dandara C, Kengne AP, Wonkam A, Okpechi I.Am J Kidney Dis. 2020 Jul;76(1):100-108. doi: 10.1053/j.ajkd.2020.01.006. Epub 2020 Apr 27.

- APOL1 risk alleles among individuals with CKD in Northern Tanzania: A pilot study. Stanifer JW, Karia F, Maro V, Kilonzo K, Qin X, Patel UD, Hauser ER.PLoS One. 2017 Jul 21;12(7):e0181811. doi: 10.1371/journal.pone.0181811.

- APOL1 genetic variants, chronic kidney diseases and hypertension in mixed ancestry South Africans. Matsha TE, Kengne AP, Masconi KL, Yako YY, Erasmus RT.BMC Genet. 2015 Jun 26;16:69. doi: 10.1186/s12863-015-0228-6.

3. The method section is not very detailed. For instance, where were the HIV negative patients recruited from and how?, little information is provided about the methods of the kidney biopsy and how it was reported, statistical methods are not sufficiently reported, etc The methods section needs to have more details to ensure clarity.

4. The results section is also lacking in detail:

- The tables do not have a legend to describe abbreviations within each Table

- Some variables do not have units (e.g. what is the unit of measurement of CD4 count?)

- Why were there two Tables labelled as “Table 3”?

- What do the empty cells in the Tables mean – “no data”, “zero”, or “not reported”?

- Why are there inconsistencies with the decimals – most of the blood pressure reported have no decimals?

- In reporting the histologies, this was stated in the results section: “…and the commonest histological diagnosis was FSGS seen in 20 (37.7%) patients followed by HIVAN seen in 19 (35.9%) …” Is this primary FSGS? How did you differentiate primary FSGS in HIV positive patients from HIVAN which also has an FSGS pattern on histology?

5. The discussion needs to probe more into the implications of your study findings, especially relative to other recent studies in the region and in Nigeria as well as implications related to kidney disease prevalence and outcomes in Nigeria.

Minor comment:

I am not sure why most of the references are very old despite there being more recent citations from the region and country.

Reviewer #3: The study investigates the frequency of high risk alleles of the APOL1 gene in HIV infected patients of Northern Nigeria and establishes their effects on kidney disease. A High Risk Genotype (HRG) is associated with an increased risk to develope HIVAN and FSGS in HIV positive group respect to the control group. In particular, carriers of 2 vs 0 risk alleles have the highest odds.

The manuscript is interesting and, despite limitations on sample size and study design that can be improved with further investigations, provides new findings on the frequencies of APOL1 genotypes in a region with limited information so far. I have some comments to clarify and better explain some matters. The PDF file doesn’t include the number of lines or the number of pages, so it was difficult to report my comments, I hope I was clear.

In the ABSTRACT, please explain the acronyms FSGS and HIVAN

In the INTRODUCTION:

1. In the phrase “these variants include the APOL1 G1 allele…..and the G2 allele” the verb “include” is not the most appropriate, the allele includes the variants and not vice-versa: please explain better the concept

2. the correct nomenclature for a protein variation is p.Ser342Gly and p.Ile384Met: please correct

3. in the phrase “such as the study among the South African black population by Kasembeli et al and another study among children in the DRC” please explain the acronym DRC

In the METHODS:

1. Which is the treatment of HIV patients (both HIV positive no CKD and HIV positive CKD)? Is there any difference between the two groups? Please explain better this subject

2. eGFR is Estimated Glomerular Filtration Rate: please correct

3. It is not specified if it has been tested the Hardy-Weinberg equilibrium: please clarify

In the RESULTS:

1. In the table 1 “Characteristics of the study participants” the unit of measure of eGFR is ml/min/1.73m2 and the unit of measure of CD4 is missing, please correct. Furthermore in my opinion it would be important to have other information such as the possible presence of coinfection (hepatitis B/C), the type of ART and the HIV viral load: please indicate whether or not you have these data, and eventually why you do not have it.

2. In the table 2 please explain what haplotypes G0, G1GM, G1G+, G1+M, G2 mean

3. In the text after table 3 you state “these associations however were not significant after adjusting for age and gender”, but the association with 2 vs 1 or 0 risk alleles is already not significant (p=0.07): please explain better this concept.

4. Why do you adjust only for age and gender? There are other risk factors that could be included as covariates in the model, such as CD4 count, viral load, ART type and blood pressure

5. There are two table 3

6. It is not specified how you determined that G1 and G2 are on two different chromosomes, therefore in compound heterozygosity (G1/G2), and not on the same: is it based on literature data or have you tested it? Please clarify this concept.

In the DISCUSSION:

1. In the phrase “variations in the frequencies of these alleles among diverse African ethnic populations…” was should be were, please correct and add the number of reference of the cited study by Tzur et al.

2. Please change the sentence “the association was also strong with HIVAN…..with a less strong association in the recessive form”, a p=0.07 is not a less strong association, is not an association at all.

3. What does it mean the expression “from this environment” in the sentence “this study provides the first information on the high frequencies of the APOL1 genotypes….”? Please explain better.

6. PLOS authors have the option to publish the peer review history of their article (what does this mean?). If published, this will include your full peer review and any attached files.

Reviewer #1: **Yes: **Himanshu Batra

Reviewer #2: No

Reviewer #3: No

---

## [Author Response · Author response to Decision Letter 0]

23 Feb 2022

Editorial concerns raised have all been addressed sequentially

Reviewer 1 All the issues raised have been addressed we are grateful to you for the valuable contributions 

Reviewer 2 We have responded to all the issues raised and we thank you

Reviewer 3. We have addressed the observations made and we thank you very much

---

## [Decision Letter · Decision Letter 1]

8 Apr 2022

PONE-D-21-10394R1HIGH RISK APOL1 GENOTYPES AND KIDNEY DISEASE AMONG TREATMENT NAÏVE HIV PATIENTS AT KANO, NIGERIAPLOS ONE

Dear Dr. Abdu,

Sorry for the long process, thank you for submitting your revised manuscript to PLOS ONE. After careful consideration, we feel that it has merit but does not fully meet PLOS ONE’s publication criteria as it currently stands. Specifically, if you could carefully response reivewer#3's comments, Therefore, we invite you to submit a revised version of the manuscript that addresses the points raised during the review process. 

We look forward to receiving your revised manuscript.

Kind regards,

Weijing He, M.D.

Academic Editor

PLOS ONE

Reviewers' comments:

Reviewer's Responses to Questions

**Comments to the Author**

1. If the authors have adequately addressed your comments raised in a previous round of review and you feel that this manuscript is now acceptable for publication, you may indicate that here to bypass the “Comments to the Author” section, enter your conflict of interest statement in the “Confidential to Editor” section, and submit your "Accept" recommendation.

Reviewer #1: All comments have been addressed

Reviewer #3: (No Response)

2. Is the manuscript technically sound, and do the data support the conclusions?

Reviewer #1: Yes

Reviewer #3: Partly

3. Has the statistical analysis been performed appropriately and rigorously? 

Reviewer #1: Yes

Reviewer #3: N/A

4. Have the authors made all data underlying the findings in their manuscript fully available?

Reviewer #1: Yes

Reviewer #3: Yes

5. Is the manuscript presented in an intelligible fashion and written in standard English?

Reviewer #1: Yes

Reviewer #3: Yes

6. Review Comments to the Author

Reviewer #1: I thank authors for addressing all the concerns which I raised. Therefore, I recommend the article to be published.

Reviewer #3: Reviewing again the manuscript, I realized that there is a serious inaccuracy in the calculation of allelic frequencies, that I have not considered before.

The 142 subjects participating in the study have 284 chromosomes and consequently alleles, so the G0 alleles in each group are not 35, 50 and 49, but 55, 88 and 76 respectively (20 individuals with the G0/G0 genotype have 40 G0 alleles; 38 G0/G0 have 76 G0 alleles; 27 G0/G0 have 54 G0 alleles); the G1 alleles are 38 and the G2 alleles are 27; in total there are 284 alleles and not 198 alleles resulting from the sum of the haplotypes listed in table 2 "Frequencies of the APOL1 risk alleles". Therefore, the sentence "The frequency of the individual risk allele among all study participants was 26.1% (37/142) for G1 and 19.0% (27/142) for G2" on page 9 is incorrect, the exact G1 prevalence is 38/284=13.4% and the G2 prevalence is 27/284=9.5%. It would be correct to consider the 142 subjects, if you are talking about the frequency of the genotype or of the individuals with a number of risk alleles of 0,1,2. If you are talking about allelic frequencies, you must consider all 284 alleles. In my opinion in a study dealing with genetics, this concept needs to be revised.

Another important issue, already included in my revision, concerns the concept on page 10, where you state "five had the compound heterozygote G1/G2 form". I would like to know how you say that, are you based on literature data or have you tested it and know that G1 and G2 are on two different chromosomes? You didn't answer to my previous question.

7. PLOS authors have the option to publish the peer review history of their article (what does this mean?). If published, this will include your full peer review and any attached files.

Reviewer #1: **Yes: **Himanshu Batra

Reviewer #3: No

---

## [Author Response · Author response to Decision Letter 1]

21 Aug 2022

1.The statement ‘the individual risk allele frequency among all study participants was’ has been changed to ‘The distribution of the APOL1 genotypes among the study participants revealed that 24.6% had a G1 risk allele and 19.0% a G2’ Page 2 line 28

2.‘The statement ‘the individual risk allele frequency among all study participants was’ has been changed to ‘The frequency of APOL 1 genotypes showed that 24.6% (35/142) of the study participants had at least 1 G1 risk allele and 19.0% (27/142) had a G2 risk allele.’ Page 9 line 164

3.The title of table 2 was changed from APOL1 risk alleles frequencies to

‘Distribution of APOL1 SNP genotypes and haplotypes in the study participants with HIV positive CKD, HIV positive no-CKD and HIV negative no-CKD’ Page 9 line 170

4.The title of table 3 was changed from APOL 1 risk alleles among the study participants to

‘APOL1 genotype distribution among HIV positive CKD patients and control patients ( HIV positive no-CKD and HIV-negative no-CKD)’ page 10 line 183

5.The statement ‘The prevalence of the individual risk alleles’ has been changed to

 ‘The prevalence of the haplotypes page 12 line 201

6.The statement ‘From the lower frequencies of these individual alleles has been changed to.

‘From the lower frequencies of these individual haplotypes’ page 13 line 214

Our response to reviewer's comment 2

This was based on the observation of the authors and published in the Kasembeli et al (2015). Because of the close proximity of the SNPs recombination is not possible/rare. The individuals showing this novel haplotype underwent several rounds of genotyping and sanger sequencing to confirm the presence of this haplotype

---

## [Decision Letter · Decision Letter 2]

28 Sep 2022

HIGH RISK APOL1 GENOTYPES AND KIDNEY DISEASE AMONG TREATMENT NAÏVE HIV PATIENTS AT KANO, NIGERIA

PONE-D-21-10394R2

Dear Dr. Aliyu Abdu,

My apologies for the long review process, we’re pleased to inform you that your manuscript has been judged scientifically suitable for publication and will be formally accepted for publication once it meets all outstanding technical requirements.

Kind regards,

Weijing He, M.D.

Academic Editor

PLOS ONE

Additional Editor Comments (optional):

Reviewers' comments:

Reviewer's Responses to Questions

**Comments to the Author**

1. If the authors have adequately addressed your comments raised in a previous round of review and you feel that this manuscript is now acceptable for publication, you may indicate that here to bypass the “Comments to the Author” section, enter your conflict of interest statement in the “Confidential to Editor” section, and submit your "Accept" recommendation.

Reviewer #1: All comments have been addressed

2. Is the manuscript technically sound, and do the data support the conclusions?

Reviewer #1: Yes

3. Has the statistical analysis been performed appropriately and rigorously? 

Reviewer #1: Yes

4. Have the authors made all data underlying the findings in their manuscript fully available?

Reviewer #1: Yes

5. Is the manuscript presented in an intelligible fashion and written in standard English?

Reviewer #1: Yes

6. Review Comments to the Author

Reviewer #1: I thank the authors for making the necessary changes.

7. PLOS authors have the option to publish the peer review history of their article (what does this mean?). If published, this will include your full peer review and any attached files.

Reviewer #1: **Yes: **Himanshu Batra

---

## [Editor Report · Acceptance letter]

3 Oct 2022

PONE-D-21-10394R2 

HIGH RISK *APOL1* GENOTYPES AND KIDNEY DISEASE AMONG TREATMENT NAÏVE HIV PATIENTS AT KANO, NIGERIA 

Dear Dr. Abdu:

I'm pleased to inform you that your manuscript has been deemed suitable for publication in PLOS ONE. Congratulations! Your manuscript is now with our production department. 

Kind regards, 

on behalf of

Dr. Weijing He 

Academic Editor

PLOS ONE